# Minimally Invasive Radical Nephroureterectomy: 5-Year Update of Techniques and Outcomes

**DOI:** 10.3390/cancers15184585

**Published:** 2023-09-15

**Authors:** Antonio Franco, Francesco Ditonno, Carol Feng, Celeste Manfredi, Morgan R. Sturgis, Mustafa Farooqi, Francesco Del Giudice, Christopher Coogan, Matteo Ferro, Chao Zhang, Zhenjie Wu, Bo Yang, Linhui Wang, Riccardo Autorino

**Affiliations:** 1Department of Urology, Rush University, Chicago, IL 60612, USA; antonio_franco@rush.edu (A.F.); francesco_ditonno@rush.edu (F.D.); carol_feng@rush.edu (C.F.); celeste_manfredi@rush.edu (C.M.); morgan_r_sturgis@rush.edu (M.R.S.); mustafafarooqi96@gmail.com (M.F.); christopher_coogan@rush.edu (C.C.); 2Department of Urology, Sant’Andrea Hospital, Sapienza University, 00189 Rome, Italy; 3Department of Urology, Azienda Ospedaliera Universitaria Integrata Verona, University of Verona, 37126 Verona, Italy; 4Urology Unit, Department of Woman, Child and General and Specialized Surgery, University of Campania “Luigi Vanvitelli”, 80138 Naples, Italy; 5Department of Maternal Infant and Urologic Sciences, Policlinico Umberto I Hospital, “Sapienza” University of Rome, 00161 Rome, Italy; francesco.delgiudice@uniroma1.it; 6Division of Urology, European Institute of Oncology (IEO), IRCCS, 20141 Milan, Italy; matteo.ferro@ieo.it; 7Department of Urology, Changhai Hospital, Naval Medical University, Shanghai 200433, China; tonyzhangchanghai@163.com (C.Z.); wuzhenjie17@163.com (Z.W.); boyangchanghai@163.com (B.Y.); wanglinhuicz@163.com (L.W.)

**Keywords:** kidney surgery, robotic urologic surgery, robot-assisted, upper tract urothelial carcinoma, ureterectomy

## Abstract

**Simple Summary:**

Minimally invasive radical nephroureterectomy is gaining momentum among upper tract urothelial carcinoma management by offering oncological radicality and less surgical morbidity. Long-term oncological outcomes suggest that it is a safe and effective treatment option for upper tract urothelial cancer.

**Abstract:**

The gold standard treatment for non-metastatic upper tract urothelial cancer (UTUC) is represented by radical nephroureterectomy (RNU). The choice of surgical technique in performing UTUC surgery continues to depend on several factors, including the location and extent of the tumor, the patient’s overall health, and very importantly, the surgeon’s skill, experience, and preference. Although open and laparoscopic approaches are well-established treatments, evidence regarding robot-assisted radical nephroureterectomy (RANU) is growing. Aim of our study was to perform a critical review on the evidence of the last 5 years regarding surgical techniques and outcomes of minimally invasive RNU, mostly focusing on RANU. Reported oncological and function outcomes suggest that minimally invasive RNU is safe and effective, showing similar survival rates compared to the open approach.

## 1. Introduction

Radical nephroureterectomy (RNU) continues to be the standard of care for upper tract urothelial carcinoma (UTUC) [1]. Historically, the procedure was performed using an open approach to access the kidney and ureter, however, there has been a major shift towards minimally invasive techniques over the past two decades [2]. First reported by Clayman in 1991 [3], laparoscopic nephroureterectomy was subsequently followed by robotic-assisted nephroureterectomy (RANU), with first case reported in 2006 [4].

Several potential benefits are associated with minimally invasive techniques, including reduced blood loss, shorter hospital stays, and faster recovery times [5]. With the introduction of the Xi platform (Intuitive Surgical, Sunnyvale, CA, USA), which is designed for “multi-quadrant” procedures, single stage RANU has been facilitated, which allowed to reduce operative time without the need to change patient’s position and/or robot’s docking. More recently, SP system hit the market [6,7] (da Vinci SP^®^ surgical system, Intuitive, Sunnyvale, CA, USA) and it might allow further advances.

Regardless of the approach, distal ureter and bladder cuff management is a fundamental step of RNU, as it highly impacts oncological results. Poorer cancer specific and overall survival were observed in patients who did not undergo complete resection of a bladder cuff [8,9]. Furthermore, several factors might impact oncological outcomes of UTUC patients undergoing RNU [10].

The aim of the present critical review is to provide a comprehensive analysis of the latest techniques and innovative approaches of minimally invasive RNU in the last 5 years, as well as the related oncologic and functional outcomes.

## 2. Literature Search Methodology

A non-systematic literature review was conducted in June 2023. PubMed and Scopus databases were explored to retrieve publications related to minimally invasive RNU from 2018 to 2023. A different combination of the following keywords was used for a title/abstract search: “nephroureterectomy”; “robotic surgery”; “robotic kidney surgery”; “robot-assisted”; “minimally invasive”; “laparoscopic”; “segmental”; “distal”; “ureter”; “ureterectomy”. Conference abstracts, review articles (except meta-analyses), editorials, commentaries, and letters to the editor were excluded from the search. Only English articles were included. Latest 5 years ‘references from selected articles were also assessed for inclusion after careful evaluation by a senior author. An evidence-based critical analysis was conducted by focusing on the latest innovative techniques described in the literature, as well as oncological and renal functional outcomes.

## 3. Surgical Techniques

### 3.1. Single Stage Robotic Radical Nephroureterectomy

RANU is a multi-quadrant surgery, which in the early robotic era demanded patient repositioning and redocking to allow access to both upper and lower urinary tract [11]. Later, investigators implemented a linear port arrangement to perform a “single stage” RANU [12]. This was initially described for the Si system [13], but it has become more established with the introduction of the Xi system (Figure 1).

In 2022 Veccia et al. [6] described a series of Xi^®^ single stage RANU in 148 patients through the ROBUUST multicenter collaborative group. Median operative time and estimated blood loss were 215.5 min and 100.0 mL, respectively; post-operative complications were 26 (17.7%) with 4 major ones (2.7%), while bladder cuff excision (BCE) and lymph node dissection were performed in 96% and 38.1% of the procedures, respectively [6]. An important aspect to be considered is the benefit and facilitation in performing one of the most challenging and fundamental RNU steps, the bladder cuff excision. In fact, a fully intracorporeal completion of this step was achieved in almost all the cases, proving that the utilization of the Xi^®^ system effectively resolves the ongoing debate on how this aspect of the procedure should be approached, thereby excising en-bloc distal ureter, ureteral-vesical junction, and bladder cuff.

### 3.2. Retroperitoneal Robotic Radical Nephroureterectomy

Despite some potential advantages of a retroperitoneal approach, this has been challenging in the case of RANU procedure, mainly because of limited working space [14,15].

Sparwasser et al. published the first series of completely retroperitoneal robot-assisted radical nephroureterectomy (RRNU) [16], and subsequently compared this technique to the standard transperitoneal approach [17]. In this scenario, ports placement starts from the Petit’s triangle and then follows a line above the iliac crest (Figure 2). As the procedure advances towards the nephrectomy stage, the robot is docked parallel to the spine with the arms pointing towards the head; after releasing the middle ureter, re-docking is performed by 180°-twist of the main joint of the robot without the need for relocation, with the trajectory of the arms towards the leg. Interestingly, given the possibility to twist and rotate the whole robot system, the authors reported only a 7 min additional time for re-docking, while most series have cited an additional 30 to 60 min [15]. Regarding BCE, only in the case of RRNU, a V-Loc (Covidien, Dublin) suture is placed at the medial dissection margin of the bladder, to prevent the potential retraction of the bladder wall prior to BCE.

Perioperative outcomes demonstrated no significant differences in terms of complications nor survival. To note, RRNU showed significantly shorter surgery time and length of stay, compared to the transperitoneal approach [17]. On the other hand, trocar placement usually requires more time than the transperitoneal approach, due to the complexity of creating the retroperitoneal working space [18].

### 3.3. Distal Ureterectomy and Bladder Cuff Excision

Various techniques have been outlined for BCE, such as open excision, transurethral resection of the ureteral orifice, ureteric intussusception, and pure laparoscopic or robotic approaches [19]. When attempting to compare outcomes between endoscopic, open or minimally invasive approaches for optimal BCE management, no clear consensus was gained [1]. However, some studies reported different findings in terms of intravesical recurrence, showing poorer outcomes for the endoscopic and laparoscopic [20,21,22,23,24].

Nevertheless, BCE by minimally invasive approaches have been implemented over the last decade. Recently, Wu et al. [25] proposed a modified retroperitoneoscopic technique that embraces the three goals for a safe and complete BCE: en-bloc excision, mucosa-to-mucosa reliable closure of the bladder opening, and no visible urine spillage. Compared to “blind” extravesical clamping techniques where essentially the distal ureter is clamped with Endo-GIA (Medtronic, Watford, UK) or Bulldogs without a proper individuation of the bladder plaque, this approach relies on maintaining tension on the ureter and meticulously incising through the bladder’s muscular layer until a substantial funnel-shaped segment of the bladder mucosa is obtained in a circumferential manner. In doing so, the distal ureter along with the bladder cuff can be easily excised en-bloc, allowing a watertight suture of the bladder defect. BCE trifecta was observed in 95% of the patients, demonstrating oncological safety of the procedure. To note, even if only one patient experienced a bladder recurrence, follow-up of the study was too short (median: 7 months) to draw substantial conclusions [25].

Worth mentioning despite the small case series, is the keyhole technique proposed by the University of Southern California team [26]. Maintaining a single position and a single docking, the distal ureter is first clipped and then the ureteral-vesical junction (UVJ) is released. A keyhole incision is performed just above the UVJ, to better identify the ureteral orifice that is subsequently excised under direct vision. In doing so, resection margins are more precisely delineated, maintaining oncologic principles of en-bloc excision without necessitating secondary cystotomy incision or concomitant endoscopic procedure. Only three patients experienced bladder recurrence and one postoperative complication was reported [26]. Again, results should be interpreted with caution due to the relatively small sample size and the lack of a control group.

### 3.4. SP Robotic Radical Nephroureterectomy

Ongoing advancements led to the introduction of the Single-Port platform (da Vinci SP^®^ surgical system, Intuitive, Sunnyvale, CA, USA). This novel platform accommodates all the robotic instruments and camera through a single multichannel 2.5 cm port inserted through a single skin incision.

To date, only a limited number of studies have documented Single-Port RNU, wherein dissection of the distal ureter and resection of the bladder cuff were conducted prior to the completion of nephrectomy, all without the need of altering the patient’s position or re-docking of the robotic system [7,27]. In fact, the da Vinci SP platform can pivot 360° around the access port, facilitating easy access to both the renal and pelvic quadrants via the same single incision.

A novel approach named SARA (Supine Anterior Retroperitoneal access) for kidney surgery, including nephroureterectomy, was recently described by Pellegrino et al. in order to gain anterior access to the retroperitoneum [7]. A 3-cm incision at the McBurney point, 3 cm medial and 3 cm caudal to the anterior superior iliac spine is performed. Subsequently, dissection of the abdominal muscles facilitates the development of the retroperitoneal space for the insertion of the da Vinci SP access port. Delicate finger dissection is then employed to carefully separate the peritoneum’s anterior reflection from the transversus abdominis muscle, creating sufficient room for the robotic access port placement (Figure 2). Advantages of the SARA technique primarily lies in the rapid access it provides to the renal hilum, as well as the easier dissection of the ureter. Regarding perioperative outcomes, the study reported a high rate of same-day discharges and a complete absence of narcotic administration, implying further potential benefits of this approach, that include reduced anesthesiologic complications thanks to the supine patient position [7].

## 4. Oncological Outcomes

Despite the above-mentioned progress in techniques, oncologic outcomes are still unsatisfactory, making UTUC a potentially deadly disease [28]. Risk of recurrence during follow-up, such as bladder, local, or distant recurrence can reach 47%, 18%, and 17% respectively [29,30], while 5-yr CSS rate are around <50% for pT2/pT3 and <10% for pT4 [31,32]. Several factors might impact the oncologic outcomes of UTUC patient, including type of treatment (open vs. minimally invasive, BCE vs. non-BCE), patient comorbidities (diabetes, acute/chronic kidney injury) and tumor features (grade, size, location, and histology) [33]. Latest oncological updates mainly involve the type of treatment used and histology variants (Table 1).

### 4.1. Bladder Cuff vs. Non-Bladder Cuff Excision

Despite recommendations from both the National Comprehensive Cancer Network (NCCN) [40] and the European Association of Urology (EAU) [41] to perform RNU with BCE, studies showed controversial results, thus increasing research focusing on this topic [36,42].

Nazzani et al. [35] questioned the effect of BCE on survival and assessed rates of guidelines adherence and implementation by investigating the Surveillance, Epidemiology, and End Results (SEER) database. Interestingly, presence or absence of BCE at RNU did not influence cancer specific mortality (CSM) or other-cause mortality (OCM). Moreover, BCE rates did show an increasing trend over time, thereby proving enhanced guidelines’ adherence in recent years.

However, as usually encountered when employing databases of this nature, information regarding type of surgical approach or BCE’ techniques, as well as features on possible chemotherapy status or cancer recurrence are missing. Nevertheless, these findings showed an encouraging improvement in guidelines’ adherence, but also revealed that more than 25% of RNUs are still performed without BCE [35].

### 4.2. Minimally Invasive vs. Open RNU

Despite the incremental diffusion of minimally invasive surgery during the last decade, controversy still exists on the differential perioperative and oncological outcomes of both robotic and laparoscopic versus open RNU [39,43,44].

According to the latest evidence, a recent systematic review of 7554 patients conducted by the EAU Guidelines panel suggests that laparoscopic bladder cuff excision appears to be associated with inferior oncologic outcomes, characterized by an increased rate of intravesical recurrence. Indeed, BCE in laparoscopic groups was performed via an open approach in most of the studies, and poorest outcomes were identified just in the former ones and in selected subgroups of patients with locally advanced (pT3/pT4) or high-grade disease, raising doubts on the importance of BCE rather than the proper type of surgical technique [45].

Regarding robotic approach, Veccia et al. [46] successfully evaluated over 87,000 RNU cases through a comprehensive and large metanalysis of 80 studies overall. Although most of each sample size was relatively small and randomized and prospective studies were lacking, results suggest that RANU appears to be a safe procedure, exhibiting the benefits of a minimally invasive approach without impairing the oncological outcomes. More specifically, when analyzing survival rates, no statistically significant differences were observed among hand-assisted laparoscopic nephroureterectomy (HALNU), laparoscopic and RANU in terms of 2- and 5-year recurrence free survival (RFS) and CSS. Noteworthy, no correlation between the surgical technique and RFS and CSS were found [46].

Notwithstanding these results, there is still an open debate regarding the best approach to adopt when dealing with locally advanced or invasive (T3/T4 and/or N+/M+) tumors. Some studies have reported atypical sites of recurrence such as port-sites metastases, peritoneal and abdominal wall implants after minimally invasive RNU [23,37]. On the other hand, cases of peritoneal cancer dissemination have been reported, but never reaching a statistical difference between the open and minimally invasive technique [47]. However, European guidelines still recommend an open approach to prevent tumor seeding in these advanced cases [38].

Despite the increasing popularity of minimally invasive RNU, persistent concerns regarding its use are pending, and the optimal surgical technique for RNU remains to be definitively established. Future clinical investigations are warranted to effectively address this issue.

### 4.3. Lymphadenectomy

The impact of Lymph Node Dissection (LND) on oncological outcomes in UTUC remains unclear [34,48]. Studies assessing the efficacy of LND during RNU, in terms of indication, extent and anatomical templates, are still controversial and debated in literature [34,41,49]. According to the latest update of the European guidelines, template based LND has a greater impact on patient survival, improving CSS and reducing the risk of local recurrence [38]. These data are further strengthened by one of the largest meta-analyses recently performed, which substantially confirmed the role of LND as a good staging procedure for UTUC disease, revealing an incidence of 13–40% of positive lymph nodes in cN0 ≥pT2 patients. Moreover, LND enhanced CSS in ≥pT2 renal pelvis tumors, thereby reducing the probability of regional lymph node metastases. However, this advantage was not evident in the case of ureteral tumors [50].

A multicenter retrospective analysis of the ROBUUST registry evaluated OS and RFS of three different cohorts who did not undergo LND (pNx), underwent LND with negative lymph nodes (pN0) and underwent LND with positive nodes (pN+), respectively. Results showed an important difference between pN+ cohort and the other two groups of patients in terms of 2 yrs OS (42% vs. 80%, 86%, *p* < 0.001) and RFS (35% vs. 53%, 61%, (*p* < 0.001). Therefore, LND during RNU in patients with positive lymph nodes provides prognostic data, but is not associated with improved OS; indeed, a poor prognosis is observed in this specific set of patients [51].

### 4.4. Impact of Histologic Variants

An additional significant factor that may influence oncological outcomes and survival rate after RNU is the presence of a histologic variant, for instance a micropapillary or sarcomatoid tumor. The incidence of such histologic variants has been documented between 7.9% to 11.8% [52].

Confirming this evidence through an extensive metanalysis, Mori et al. demonstrated a significant correlation with unfavorable outcomes for variant histology, in terms of CSS, OS and RFS. A subsequent subgroup analysis further revealed that specific variant histology, such as micropapillary and squamous and/or glandular variants, were particularly associated with poorer CSS [53].

A multi-institutional study conducted by the ROBUUST collaborative group evaluated the impact of histologic variants on oncological outcomes in patients who underwent RANU. According to the literature’ incidence, the most common variant encountered was squamous followed by micropapillary and sarcomatoid, within a total of 70 patients out of 687 (10.2%). Oncologic outcomes revealed an increased risk of metastasis and death for patients with these variants. However, on multivariable analysis, OS rates and the risk of urothelial recurrence in the bladder or contralateral kidney were not affected by the presence of histologic variants [54].

Furthermore, RNU outcomes following a diagnosis of primary or concomitant carcinoma in situ (CIS) have been poorly explored. For this reason, the Nishinihon Uro-Oncology Collaborative Group [55] first attempted to compare prognostic features between primary and concomitant CIS in a multicenter study. Within a cohort of 163 patients diagnosed with either primary or concomitant CIS following RNU, intriguingly, they discovered that 10 yrs CSS was significantly longer in patients with pure/primary CIS rather than in concomitant CIS ones (111.8 vs. 85.89 months).

The current analysis represents the first description of the natural course of primary CIS in the upper tract managed by surgery [55]. According to the following outcomes, concomitant CIS in the upper tract might be a potential marker of aggressive alterations and therefore, patients presenting with such histology may benefit from multimodal therapeutic approaches, including the possibility of neoadjuvant or adjuvant chemotherapy.

## 5. Renal Functional Outcomes

Besides cancer control, preservation of renal function is one of the primary goals of UTUC management. Achieving renal function preservation in patient who underwent RNU can be notably challenging due to several factors, including prevalence of chronic kidney disease (CKD), renal associated comorbidities (hypertension, diabetes) and cisplatin-based chemotherapy, which is also an important consideration for patients with high-risk tumors [56,57].

Recent studies have investigated the role of renal function variation after RNU with the aim of predicting renal function recovery, to better counsel patient candidate to adjuvant treatment [58,59]. Dividing a cohort of patients undergoing RNU in relation to their eGFR, Lee et al. showed that cumulative incidence of eGFR recovery was significantly higher in patients with low baseline eGFR (≤60 mL/min) compared to those with high baseline eGFR (≥60), with recovery rates at 2 years of 56.6% and 27.7%, respectively. Interestingly, on multivariable analysis both preoperative hydronephrosis and eGFR ≤60 were significant predictors of renal function recovery [60].

These findings were partially confirmed later by a multicenter study conducted by the RaNeO research consortium [61], where the presence of hydronephrosis was associated with lower renal function reduction. A possible explanation of this phenomenon could rely on the fact that established contralateral compensatory kidney hypertrophy of the ipsilateral urinary tract facilitates the compensatory role of the remnant solitary kidney.

On the other hand, recent evidence suggests that preoperative eGFR ≤60 might have a negative impact on renal function recovery [62,63]. Moreover, a detrimental effect of postoperative acute kidney injury on eGFR can still be recognized at 6 and 12 months after surgery [61].

Notably, a nomogram predicting renal insufficiency for cisplatin-based adjuvant chemotherapy after minimally invasive RNU was developed. Including age, BMI, preoperative eGFR and hydronephrosis, this tool showed an accuracy of 77% after external validation, further implemented by dividing the cohort in low-risk and high-risk patients. In doing so, this prognostic tool might help in the discernment of treatment options in UTUC patients [64].

As a matter of fact, these results may prove important clinical implications: in the context of radical surgery as RNU, timely detection of patients who are at major risk of experiencing a reduction in eGFR and are no longer suitable candidates for adjuvant therapy, may take advantage from neoadjuvant treatment strategies, resulting in survival’s increase. Conversely, patients who are ineligible for neoadjuvant therapy face an elevated risk of encountering a decline in renal function after RNU. For such individuals, kidney-sparing surgical interventions may be suggested, as they can mitigate the morbidity associated with radical surgery while preserving acceptable oncological outcomes.

## 6. Future Perspectives

As we continue to advance our understanding of UTUC management, several key areas of research and innovation emerge as critical for the future. These directions aim to further improve patient outcomes, refine surgical techniques, and enhance our understanding of the disease.

One of the most promising avenues for future research in UTUC is the development of precision medicine approaches. Identifying specific biomarkers that can predict treatment response and prognosis is crucial. Genomic and molecular profiling of UTUC tumors may help tailor treatments, such as targeted therapies or immunotherapies, to individual patients, especially the ones affected by Lynch syndrome [65,66].

Notably, variations in microsatellite instability (MSI) 678 frequency and hypermethylation status have been documented between UTUC and bladder urothelial carcinoma (BUC) [67,68]. Patients with Lynch syndrome face an elevated risk of developing UTUC more often than BUC when compared to the general population [66]. These distinctions could potentially offer additional prospects for clinical advantages from immune-checkpoint inhibitor therapy in a select group of individuals with MSI.

Furthermore, higher incidence of FGFR3 mutations in UTUC compared to BUC have been reported by earlier investigations [69,70]. This might be related to biological differences between the two types of urothelial cancer. In fact, UTUC more frequently exhibits gene expression patterns consistent with a luminal urothelial carcinoma molecular subtype, while BUC tends to express genes associated with urothelial basal cells and the basal-like subtype [65,69]. These biological distinctions may potentially impact the response to immune-checkpoint inhibition therapy and warrant the need of distinct clinical trials involving targeted therapies.

Another potential tool that is certainly crucial in UTUC treatment strategy, as suggested and confirmed by European Guidelines, is the use of prognostic models [41]. Among them, nomograms may serve as a user-friendly instrument for estimating an individual patient’s risk of experiencing a particular event, such as tumor recurrence or mortality [71]. For instance, evaluating the risk before surgery aims to determine the most appropriate treatment approach for patients with localized disease: kidney sparing surgery for low-risk and radical nephroureterectomy for high-risk patients [72]. Furthermore, as previously reported, postoperative risk stratification may help deciding the administration of adjuvant chemotherapy and better defining the follow-up strategy [64]. Over the past decades, various nomograms have been developed for postoperative UTUC patient counseling [73]. However, there is still a lack of knowledge on the practicality and accuracy of these tools, with particular concern about their limited use in routine clinical practice.

Finally, collaboration between urologists, oncologists, pathologists, radiologists, and other experts is vital for advancing UTUC research. Multidisciplinary tumor boards should be established to discuss complex cases and develop personalized treatment plans. These collaborations can facilitate the translation of research findings into clinical practice.

## 7. Conclusions

Minimally invasive techniques have become well-established in the management of UTUC. RANU is rapidly becoming the new standard for minimally invasive RNU in many Centers. Both the transperitoneal and retroperitoneal approaches were shown to be effective and feasible, equally maintaining surgical radicality and safeness, although the choice of BCE technique remains key to maximize oncological results. Preserving renal function is mandatory since is the most common cause of cisplatin-based treatment ineligibility; therefore, the availability of predictive tools for assessing renal functions’ decline should optimize perioperative management planning and helps in the identification of patients who most likely would benefit from neoadjuvant chemotherapy.

## Figures and Tables

**Figure 1 cancers-15-04585-f001:**
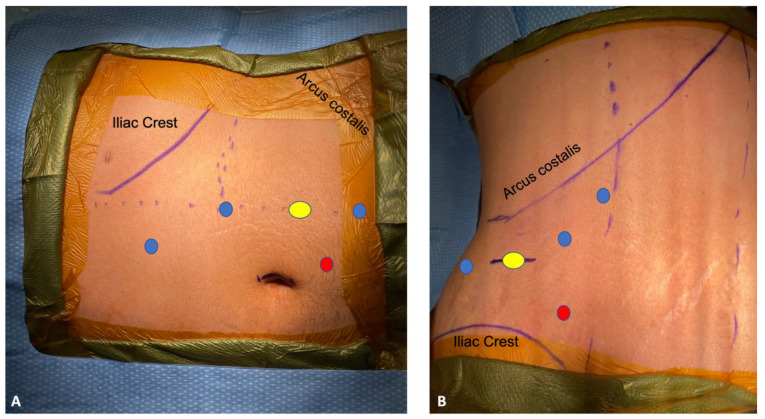
Robotic Nephroureterectomy (RANU) ports placement. (**A**) Single stage Transperitoneal RANU; (**B**) Retroperitoneal RANU, Blu circle: 8 mm Robotic Port; Yellow circle: 8 mm Camera Port; Red Circle: 12 mm Assistant Port.

**Figure 2 cancers-15-04585-f002:**
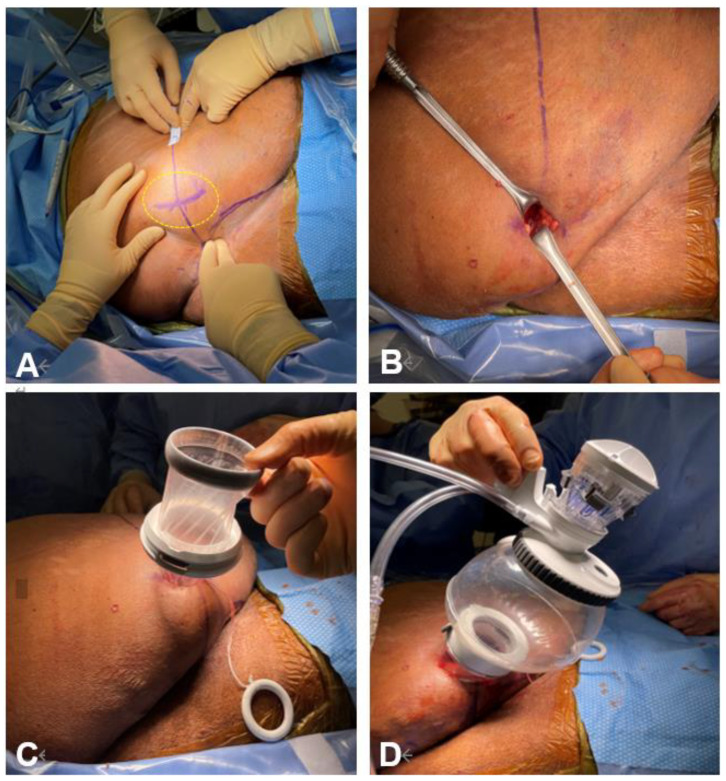
SARA access. (**A**) Incision site at McBurney point, 3 cm medial and 3 cm caudal to the anterior superior iliac spine; (**B**) 3 cm incision; (**C**) Wound retractor insertion; (**D**) SP access port placement. SARA: Supine Anterior Retroperitoneal Access; SP: Single Port.

**Table 1 cancers-15-04585-t001:** Oncological outcomes of Radical Nephroureterectomy: literature overview.

Study Name	Year	Type of Study	*N* of Cases	Topic	Main Results
Inamoto [34]	2018	Retrospective two-arm comparative study	163	Variant Histology p-CIS vs. c-CIS	10 yrs CSS: p-CIS 111.8 months c-CIS 85.9 months
Upfill-Brown [35]	2019	Retrospective two-arm comparative study (NCDB Database)	16,783	Nephroureterectomy vs. Endoscopic Management	ET worse OS vs. RNU (HR 1.43; *p* = 0.006)
Nazzani [36]	2020	Retrospective two-arm comparative study (SEER Database)	4266	RNU + BCE vs. RNU	5 yrs CSM: BCE 19.7% vs. No BCE 23.5% (*p* = 0.005) ± BCE (HR 1.14; *p* = 0.1)
Peyronnet [22]	2019	Meta-analysis	7554	Laparoscopic vs. Open RNU	CSS, RFS, MFS: *p* = 0.2, *p* = 0.86, and *p* = 0.12 pT3/HG Open vs. Lap (*p* < 0.05)
Veccia [37]	2020	Meta-analysis	87,291	Robotic vs. Lap vs. Open RNU	RANU vs. Lap vs. Open RFS: 0.99; CSS: 0.83
Mori [38]	2020	Meta-analysis	12,865	Variant Histology	CSS: HR 2.00 OS: HR 1.76 RFS: HR 1.64
Kawada [39]	2023	Meta-analysis	N/A	Nephroureterectomy vs. Endoscopic Management	OS: HR 1.27 CSS: HR 1.37

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
