# Peer review of "Minimally Invasive Radical Nephroureterectomy: 5-Year Update of Techniques and Outcomes"

_cancers, 2023, doi:10.3390/cancers15184585_

Round 1
Reviewer 1 Report
The authors provide a review of minimally-invasive nephroureterectomy. The review is very well organized and contains very useful information for urologists. Further improvement would be desirable by considering the following points
1 ) Several guidelines mention the importance of lymph node dissection for upper tract urothelial carcinoma; how about summarizing and describing the current status of lymph node dissection in minimally invasive nephroureterectomy? What do you think?
2) There is concern about atypical recurrence in minimally-invasive surgery. What are the reports on minimally-invasive nephroureterectomy?
Author Response
1) Several guidelines mention the importance of lymph node dissection for upper tract urothelial carcinoma; how about summarizing and describing the current status of lymph node dissection in minimally invasive nephroureterectomy? What do you think?
Thanks for your comment. We agree this is an important topic. However, please note that the issue of lymph node dissection will be addressed in a separate manuscript of this special issue. Nevertheless, we decided to add a brief section on the topic in this revised draft.
“3.2.3. Lymphadenectomy
The impact of Lymph Node Dissection (LND) on oncological outcomes in UTUC remains unclear47,48. Studies assessing the efficacy of LND during RNU, in terms of indication, extent and anatomical templates, are still controversial and debated in literature35,47,49. According to the latest update of the European guidelines, template-based LND has a greater impact on patient survival, improving CSS and reducing the risk of local recurrence46. These data are further strengthened by one of the largest meta-analyses recently performed, which confirmed the role of LND as a good staging procedure for UTUC disease, revealing an incidence of 13-40% of positive lymph nodes in cN0 ≥ pT2 patients. Moreover, LND enhanced CSS in ≥pT2 renal pelvis tumors, thereby reducing the probability of regional lymph node metastases. However, this advantage was not evident in the case of ureteral tumors50.
A multicenter retrospective analysis of the ROBUUST registry evaluated OS and RFS of three different cohorts who did not undergo LND (pNx), underwent LND with negative lymph nodes (pN0) and underwent LND with positive nodes (pN+), respectively. Results showed an important difference between pN+ cohort and the other two groups of patients in terms of 2 yrs OS (42% vs 80%, 86%, p <0.001) and RFS (35% vs 53%, 61%, (p <0.001). Therefore, LND during RNU in patients with positive lymph nodes provides prognostic data, but is not associated with improved OS; indeed, a poor prognosis is observed in this specific set of patients51”.
2) There is concern about atypical recurrence in minimally-invasive surgery. What are the reports on minimally-invasive nephroureterectomy?
Yes, we do agree this is also an important issue. We added the following paragraph:
“Notwithstanding these results, there is still an open debate regarding the best approach to adopt when dealing with locally advanced or invasive (T3/T4 and/or N+/M+) tumors. Some studies have reported atypical sites of recurrence such as port-sites metastases, peritoneal and abdominal wall implants after minimally invasive RNU23,44. On the other hand, cases of peritoneal cancer dissemination have been reported, but never reaching a statistical difference between the open and minimally invasive technique45. However, European guidelines still recommend an open approach to prevent tumor seeding in these advanced cases46”.
Reviewer 2 Report
Major comments: Sufficient description as a review article.
Minor comments:
1. Is Reference No. 6 listed in the correct manner?
2. The number of authors and other information in the references are listed in different ways.
3. From which literature does Figure 1 cite the figure? The figure differs from the literature cited.
4. Line 100 (Figure 2). When approaching Is this correct?
5. Line 226 sill remain Is this correct?
6. Line 262 including prevalence of CDK, Is this correct?
7. Where is Figure 2 also taken from? Or is it created by the author?
Author Response
The authors thank the reviewer for his/her great comments.
We have revised all the grammar and spelling mistakes, as well as references’ list according to the reviewer's suggestions. We have further implemented the discussion on several topics to extend the wideness of our manuscript, according also to other reviewer’ suggestion.
You can find our manuscript’s corrections highlighted in yellow in the revised draft.
- Is Reference No. 6 listed in the correct manner?
Yes
- The number of authors and other information in the references are listed in different ways.
This was corrected.
- From which literature does Figure 1 cite the figure? The figure differs from the literature cited. 7. Where is Figure 2 also taken from? Or is it created by the author?
Thank you for your feedback. All the images are property of the authors. We had previously referenced them because they were explicative of approaches described in other studies. According to your suggestion, we have removed linked references.
- Line 100 (Figure 2).When approaching Is this correct?
We have corrected: “As the procedure advances towards the nephrectomy stage”
- Line 226 sill remain Is this correct?
We have corrected: “still”
- Line 262 including prevalence of CDK, Is this correct?
We have corrected: “chronic kidney disease”
Reviewer 3 Report
please declare contributions that the authors made in this paper,
as it associate with many people from different institutions.
Author Response
The authors thank the reviewer for his/her suggestion. We have inserted an author contributions’ statement in the manuscript:
“Author Contributions: “Conceptualization, A.F. and C.F.; methodology, F.D. and C.M.; software, C.Z.; validation, Z.W., M.F. and C.C.; investigation, F.DG.; resources, B.Y.; data curation, C.M.; writing—original draft preparation, A.F.; writing—review and editing, A.F. and M.F.; visualization, M.S.; supervision, R.A.; project administration, L.W. and R.A. All authors have read and agreed to the published version of the manuscript.”